# Effect of stimulation time on the expression of human macrophage polarization markers

**Duygu Unuvar Purcu**[1,2], **Asli Korkmaz**[1,3☯], **Sinem Gunalp**[1,3☯], **Derya Goksu Helvaci**[4], **Yonca Erdal**[1], **Yavuz Dogan**[5], **Asli Suner**[6], **Gerhard Wingender**[1,7], **Duygu Sag**[1,3,8]*

**1** Izmir Biomedicine and Genome Center, Izmir, Turkey, **2** Department of Molecular Medicine, Health Sciences Institute, Dokuz Eylul University, Izmir, Turkey, **3** Department of Genomic Sciences and Molecular Biotechnology, Izmir International Biomedicine and Genome Institute, Dokuz Eylul University, Izmir, Turkey, **4** Faculty of Medicine, Dokuz Eylul University, Izmir, Turkey, **5** Department of Microbiology, Faculty of Medicine, Dokuz Eylul University, Izmir, Turkey, **6** Department of Biostatistics and Medical Informatics, Faculty of Medicine, Ege University, Izmir, Turkey, **7** Department of Biomedicine and Health Technologies, Izmir International Biomedicine and Genome Institute, Dokuz Eylul University, Izmir, Turkey, **8** Department of Medical Biology, Faculty of Medicine, Dokuz Eylul University, Izmir, Turkey

☯ These authors contributed equally to this work.
* duygu.sag@ibg.edu.tr

**Data Availability Statement:** All relevant data are within the manuscript and its Supporting Information files.

**Funding:** This work was funded by the Co-Funded Brain Circulation Scheme (MSCA/TUBITAK,

## Abstract

Macrophages are highly plastic cells that can polarize into functionally distinct subsets *in vivo* and *in vitro* in response to environmental signals. The development of protocols to model macrophage polarization *in vitro* greatly contributes to our understanding of macrophage biology. Macrophages are divided into two main groups: Pro-inflammatory M1 macrophages (classically activated) and anti-inflammatory M2 macrophages (alternatively activated), based on several key surface markers and the production of inflammatory mediators. However, the expression of these common macrophage polarization markers is greatly affected by the stimulation time used. Unfortunately, there is no consensus yet regarding the optimal stimulation times for particular macrophage polarization markers in *in vitro* experiments. This situation is problematic, (i) as analysing a particular marker at a suboptimal time point can lead to false-negative results, and (ii) as it clearly impedes the comparison of different studies. Using human monocyte-derived macrophages (MDMs) *in vitro*, we analysed how the expression of the main polarization markers for M1 (CD64, CD86, CXCL9, CXCL10, HLA-DR, IDO1, IL1β, IL12, TNF), M2a (CD200R, CD206, CCL17, CCL22, IL-10, TGM2), and M2c (CD163, IL-10, TGFβ) macrophages changes over time at mRNA and protein levels. Our data establish the most appropriate stimulation time for the analysis of the expression of human macrophage polarization markers *in vitro*. Providing such a reference guide will likely facilitate the investigation of macrophage polarization and its reproducibility.

## Introduction

Macrophages are phagocytes that play essential roles in both innate and adaptive immunity and are involved, for example, in host defence, regulation of metabolism, and tissue

#115C074), www.tubitak.gov.tr; the Dokuz Eylul University (#2017.KB.SAG.011), https://www.deu.edu.tr; the Turkish Academy of Sciences (TUBA-GEBIP award), http://www.tuba.gov.tr/tr/; and the Science Academy, Turkey (BAGEP award), https://bilimakademisi.org/ (all to D.S). The funders had no role in study design, data collection and analysis, decision to publish, or preparation of the manuscript.

**Competing interests:** The authors have declared that no competing interests exist.

remodelling [1–3]. An essential feature of macrophages is their plasticity, whereby varying environmental signals can induce the differentiation of macrophages into functional distinct subsets [4–6]. Mirroring the Th1/Th2 paradigm, macrophages can polarize into classically activated M1 or alternatively activated M2 phenotypes [4, 7, 8]. Although the M1/M2 dichotomy appears to be a simplification of the *in vivo* situation, it is a helpful experimental distinction that greatly contributed to our understanding of macrophage biology [9].

M1 macrophages are proinflammatory cells that display anti-tumour and anti-microbial activities [10, 11]. M1 activation is induced by IFNγ alone or in combination with LPS or TNF [8, 12]. M1 macrophages secrete large amounts of pro-inflammatory cytokines such as TNF, IL-1β, IL-6, IL-12, and IL-23, as well as chemokines like CXCL9 and CXCL10. They also express MHC class II (e.g. HLA-DR in humans) and costimulatory molecules including CD80 and CD86 [13–15]. In addition, M1 macrophages upregulate the expression of the tryptophan catabolizing enzyme IDO1 and of SOCS3, a suppressor of M2 cytokine signaling [8, 16].

M2 macrophages are anti-inflammatory cells that play a role in angiogenesis, wound healing, and tissue remodelling [17, 18]. M2 macrophages can further be divided into M2a, M2b, M2c, and M2d subtypes [14, 19, 20]. M2a macrophages can be induced by IL-4 and/or IL-13 and they express CD206 (MRC1), CD200R, the decoy receptor IL-1RII, TGM2 (transglutaminase 2) and the chemokines CCL17, CCL22 and CCL24. M2b macrophages can be induced by immune complexes, and TLR agonists, or IL-1 receptor ligands and they can express IL-6, IL-10, IL-1β, and TNF [12, 20, 21]. M2c macrophages can be induced by IL-10, glucocorticoids, or TGFβ and express CD163, IL-10, and TGFβ [12, 22]. Finally, M2d macrophages can be induced by TLR agonists or adenosine receptor activation and they express IL-10, IL-12, TGFβ, and TNF [23, 24].

Significant differences between mice and humans for the expression of macrophage polarization markers have been noted. For instance, while murine M1 macrophages express inducible NO synthase (iNOS), which is involved in L-arginine metabolism, human M1 macrophages do not [25]. Moreover, murine M2a macrophages induced by IL-4 and/or IL-13 express Arg1, Ym1, and Fizz1, but these genes do not have human homologs [14, 19, 26, 27]. Due to these differences, it is unclear to which extent data on murine M1/M2 polarization mimic human macrophage biology. Although monocytic cell lines such as THP-1 are commonly used for human macrophage studies, these cells display some substantial differences compared to primary human macrophages [28]. For example, PMA-treated THP-1 macrophage-like cells do not express HLA-DR and CD206 but show higher expression of CD14 and IL-1β compared to primary human macrophages [29]. Furthermore, it was reported that the response to LPS stimulation by THP-1 cells is weaker than by primary cells [30] and the expression of CD14 and cytokines by THP-1 cells was sensitive to the culture conditions such as the cell density [31]. Therefore, experiments using primary human macrophages are most suitable to understand macrophage polarization in humans.

While there is largely agreement on which markers are best for the analysis of macrophage polarization [8, 15, 24, 32], this is not the case for the optimal time point to do so. However, the consideration of the stimulation time is important as the expression of common macrophage polarization markers greatly changes over time. This leads to the unfortunate situation that many studies report on the inability of a particular macrophage subset to respond, that may well be false negative data as a suboptimal time point for the analysis was chosen. This lack of consensus on the optimal stimulation time of each polarization marker, greatly complicates the comparison of different studies and their findings. However, to our best knowledge, no comprehensive time course analysis of the common polarization markers has been published. Here we report a detailed time course of the expression of the main M1, M2a, and M2c polarization markers of human monocyte-derived macrophages (MDMs) in vitro at both

mRNA and protein levels. Our data establish the optimal time points for the analysis of the expression of these polarization markers, which will greatly facilitate their investigation.

## Materials and methods

### Generation of human monocyte-derived macrophages (MDMs)

The ethical approval for buffy coats was obtained from the "Non-Interventional Research Ethics Committee" of the Dokuz Eylül University (Approval number: 2019/02-39). The buffy coats from healthy donors were obtained from the Dokuz Eylul University Blood Bank (Izmir, Turkey) after written consent. Human monocytes were purified from buffy coats by two gradient centrifugations: PBMCs were initially isolated with Ficoll-Paque (GE Healthcare, Pittsburgh, PA) and subsequently peripheral monocytes were isolated with Percoll (GE Healthcare, Pittsburgh, PA) [33]. To generate human monocyte-derived macrophages (MDMs), monocytes were treated for 7 days with 10 ng/mL human recombinant M-CSF (PeproTech, Rocky Hill, NJ) in R5 medium (RPMI 1640 (Gibco, Thermo Fisher Scientific, Waltham, MA, USA) supplemented with 5% heat inactivated fetal bovine serum (Gibco, Thermo Fisher Scientific, Waltham, MA, USA), 1% Penicillin-Streptomycin (Gibco, Thermo Fisher Scientific, Waltham, MA, USA)) in ultra-low attachment six-well plates (Corning Life Sciences, Tewksbury, MA) [34]. On day 7, the mature macrophages were collected from the low attachment plates and verified to be more than 90% CD68[+] by flow cytometry.

### Polarization of human MDMs

Human MDMs were plated in 24 well plates at 6.5 x $10^5$ cells per well in 1 mL R5 medium and allowed to rest overnight at 37°C before stimulation. Then, the human MDMs (M0 macrophages) were either left unstimulated or stimulated for 4h, 8h, 12h, 24h, 48h, or 72 h to induce macrophage polarization: (i) for M1 polarization with 100 ng/mL LPS (Ultrapure; InvivoGen, San Diego, CA) and 20 ng/mL IFNγ (R&D, Minneapolis, MN), (ii) for M2a polarization with 20 ng/mL IL-4 (R&D, Minneapolis, MN), or (iii) for M2c polarization with 20 ng/mL IL-10 (R&D, Minneapolis, MN). The viability of the LPS+ IFNγ treated human MDMs was analyzed by flow cytometry and was over 92% for all time points.

### ELISA

After stimulations, supernatants were collected and TNF (Invitrogen, Thermo Fisher Scientific, Waltham, MA, USA), IL-12 (BioLegend, San Diego, CA), IL-10 (BioLegend, San Diego, CA), IL-1β (BioLegend, San Diego, CA), and TGFβ (R&D, Minneapolis, MN) concentrations were determined by ELISA according to the manufacturer's instructions.

### Quantitative real time PCR

Total RNA of macrophages was isolated using the Nucleo-Spin RNA kit (Macharey Nagel, Germany). RNA purity and quantity were evaluated with Nanodrop spectrophotometer (Thermo Scientific, Rockford, IL). cDNA synthesis was performed using EvoScript Universal cDNA Master, using 1000 ng of RNA per sample (Roche, Basel, Switzerland). Quantitative PCR (qPCR) was performed with a LightCycler 480 II real time system (Roche, Basel, Switzerland) using FastStart Essential DNA Probes Master (Roche, Basel, Switzerland) and the following RealTime ready Single Assays (Roche, Basel, Switzerland) for IDO1 (100134768), CXCL10 (100134759), TNF (100134777), CXCL9 (100137998), MRC1 (100134731), TGM2 (100134722), CCL17 (100138007), CCL22 (100134713), IL-10 (100133437), CD163 (100134801), and ACTB (100074091). qPCR for the markers below was performed with

Applied Biosystems™ 7500 Real-Time PCR System (Waltham, Massachusetts, USA) using FastStart Essential DNA Green Master (Roche, Basel, Switzerland) and the following QuantiTect Primer Assay (Qiagen, Düsseldorf, Germany) for IL1β (QT00021385), IL12A (QT00000357), IL-10 (QT00041685), TGFβ (QT00000728) and ACTB (QT00095431). RNA isolation, cDNA synthesis, and the quantitative PCR were performed according to the manufacturers' recommendations. Relative mRNA expression was calculated using the $2^{-\Delta\Delta Ct}$ method [35] and β-actin was used for normalization as housekeeping gene.

## Flow cytometry

Macrophages were detached from the cell plate surface using StemPro Accutase Cell Dissociation Reagent (Gibco; Thermo Fisher Scientific, Waltham, MA, USA) according to manufacturer's instructions. Single-cell suspensions were stained in flow cytometry staining buffer (PBS, 1% bovine serum albumin, 0.1% sodium azide). Zombie UV Fixable Viability Kit (BioLegend, San Diego, CA, USA) was used to determine cell viability. Fcγ receptors were blocked with Human TruStain FcX antibody (BioLegend, San Diego, CA) for 10 min and surface antigens on cells were stained for 40 min at 4˚C. Following anti-human antibodies were used, with their fluorochrome, clone, and dilution: CD86-BV605 (IT2.2; 1:400), HLA-DR-APC-Cy7 (L243; 1:400), CD64-PerCP-Cy5.5 (10.1; 1:200), CD200R-PE-Dazzle594 (OX-108; 1:400), CD206-AF700 (15–2; 1:400), CD163-PE-Cy7 (GHI/61; 1:200). Antibodies were purchased from BioLegend (San Diego, CA, USA). Cell fluorescence was measured using an LSR Fortessa (BD Biosciences) and the data were analyzed with FlowJo software (TreeStar, Ashland, OR). Forward- and side-scatter parameters were used to exclude doublets from the analysis.

## Statistical analysis

Results are expressed as mean ± SEM. A repeated measures ANOVA with a post hoc test, Least Significant Difference (LSD) adjustment, was used to compare group means of experimental data at each time point. The p-values were calculated by two-tailed and less than 0.05 was considered statistically significant. Statistical analyses were performed using IBM SPSS version 25.0 statistical software. All graphs were generated using Graph Pad Prism 8 (GraphPad Software, CA, USA).

## Results

### Time course dependent changes in the expression of polarization markers at mRNA level in M1, M2a, and M2c macrophages

To establish the optimum stimulation times for the expression of the common markers used to identify M1/M2 macrophage subtypes [20, 32, 36–38], primary human monocyte-drived macrophages (MDMs) were first analyzed at the mRNA level. Primary human MDMs were polarized into an M1 (100 ng/mL LPS, 20 ng/mL IFNγ), an M2a (20 ng/mL IL-4), or an M2c (20 ng/mL IL-10) phenotype for 4, 8, 12, 24, 48, or 72 h. The expression of M1, M2a, or M2c markers was analyzed at mRNA level by qPCR. Unpolarized human MDMs were used as control. In M1 macrophages, the mRNA expression of the M1 markers CXCL9, CXCL10, and TNF reached their highest level after 4 h of stimulation with a gradually decrease thereafter (Fig 1A–1C). The expression of IL-1β sigificantly increased at 4 h of stimulation and the level of expression stayed stable at later time points (Fig 1D). Unlike the other M1 cytokines, IL-12 gene expression decreased below basal level after 4 h and significantly increased starting 48 h (Fig 1E). In contrast, the IDO1 expression in M1 macrophages continuously increased during the time course reaching its highest value at 48–72 h (Fig 1F).

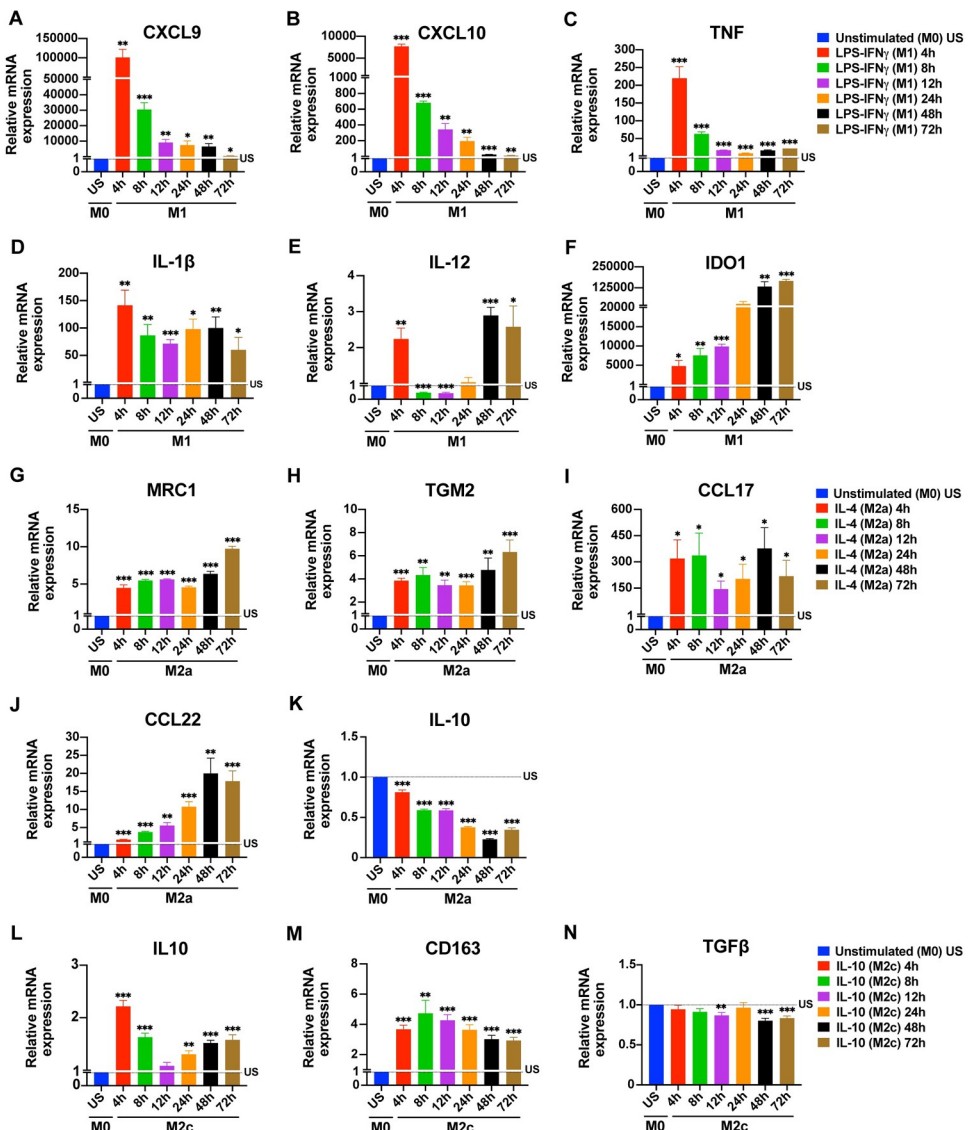

**Fig 1. Time depended changes in the expression of macrophage polarization markers at mRNA level.** Primary human MDMs (M0) were left unstimulated or were polarized to M1 (100 ng/mL LPS, 20 ng/mL IFNγ), M2a (20 ng/mL IL-4), or M2c (20 ng/mL IL-10) for 4, 8, 12, 24, 48, or 72 h. The expression of **(A-F)** CXCL9, CXCL10, TNF, IL-1β, IL-12, and IDO1 mRNA for M1 polarization; of **(G-K)** MRC1, TGM2, CCL17, CCL22 and IL-10 mRNA for M2a polarization; and of **(L-N)** IL-10, CD163 and TGFβ mRNA for M2c polarization were analyzed by qPCR. Data shown are mean ± SEM of biological replicates of 3 independent donors. Polarized macrophages (M1, M2a, or M2c) at all time points were compared with unstimulated (US) M0 macrophages. Trends for individual donors are depicted in S1 Fig. Statistical analyses were performed with repeated measures ANOVA, $^*$ $p < 0.05$; $^{**}$ $p < 0.01$; $^{***}$ $p < 0.001$ (Statistical comparison of all time points are shown in the S1 Table).

In M2a macrophages, the M2a markers MRC1 (CD206) and TGM2 were induced after 4 h without major changes until the 48 h time point. A further increase for both was noted at 72 h (Fig 1G and 1H). Compared to the other M2a markers, CCL17 expression was very high at all time points (Fig 1I). The expression of CCL22 in M2a macrophages gradually increased from 4h to 48h (Fig 1J). Surprisingly, IL-10 expression gradually decreased below baseline between 4–72 h (Fig 1K). In M2c macrophages, the M2c markers IL-10, CD163 and TGFβ were analyzed. The expression of IL-10 mRNA peaked at 4h (Fig 1L), whereas the expression of CD163

**Table 1. Suggested stimulation times for the detection of M1, M2a, and M2c polarization markers in human monocyte-derived macrophages.**

| Macrophage Phenotype | Stimulant | Detection Method | Marker | Type | Suggested Stimulation Time for Optimal Detection |
|---|---|---|---|---|---|
| **M1** | LPS + IFNγ | qPCR | **CXCL9** | Chemokine | 4–8 h |
| | | | **CXCL10** | Chemokine | 4–8 h |
| | | | **TNF** | Cytokine | 4–8 h |
| | | | **IL-1β** | Cytokine | 4–72 h |
| | | | **IL-12** | Cytokine | 4 h, 48–72 h |
| | | | **IDO1** | Enzyme | 4–72 h |
| | | Flow Cyt. | **CD86** | Co-stimulatory M. | 8–12 h |
| | | | **CD64** | Fc Receptor | 24–72 h |
| | | | **HLA-DR** | MHC class II M. | 12 h |
| | | ELISA | **TNF** | Cytokine | 4–72 h |
| | | | **IL12p70** | Cytokine | 8–72 h |
| | | | **IL-1β** | Cytokine | 8–24 h |
| **M2a** | IL-4 | qPCR | **MRC1** | Mannose R. | 4–72 h |
| | | | **TGM2** | Enzyme | 4–72 h |
| | | | **CCL17** | Chemokine | 4–72 h |
| | | | **CCL22** | Chemokine | 8–72 h |
| | | | **IL-10** | Cytokine | nd |
| | | Flow Cyt. | **CD200R** | Inhibitory R. | 24–72 h |
| | | | **CD206** | Mannose R. | 24–72 h |
| | | ELISA | **IL-10** | Cytokine | 48–72 h |
| **M2c** | IL-10 | qPCR | **IL-10** | Cytokine | 4 h |
| | | | **CD163** | Scavenger R. | 4–72 h |
| | | | **TGFβ** | Cytokine | nd |
| | | Flow Cyt. | **CD163** | Scavenger R. | 24–72 h |
| | | ELISA | **TGFβ** | Cytokine | 72h |

Flow Cyt., flow cytometry; R, receptor; P, protein; M, molecule; nd, no significant upregulation was detected.

mRNA was comparable between 4 h and 72 h (Fig 1M). We did not observe TGFβ gene expression at any time points analyzed (Fig 1N).

As donor-donor variability in human macrophages has been reported [39, 40], we also graphed the changes of the expression of the analyzed polarization markers over time for each donor separately (S1–S3 Figs) to illustrate the observed variability.

These data indicate that 4–8 h is the optimal stimulation time for the simultaneous detection of the M1 markers, at the mRNA level. While for M2c markers, except TGFβ, 4 h of stimulation seems ideal; M2a markers, except IL-10, can be simultaneously detected between 8–72 h of stimulation (Table 1).

## Time course depended changes in the expression of surface markers at protein level in M1, M2a and M2c macrophages

Next, to determine the impact of stimulation time on the expression of surface markers, the primary human MDMs were analyzed for the expression of M1, M2a or M2c surface markers by flow cytometry (Fig 2, S2 Fig). In human MDMs polarized into an M1 phenotype, the CD86 expression was, surprisingly, bimodal, with peaks after 12 h and 72 h of stimulation and at background levels at the 24 h time point (Fig 2A). The CD64 expression was increased from 8 h to 72 h (Fig 2B). The expression of HLA-DR was significantly enhanced only at 12 h (Fig 2C).

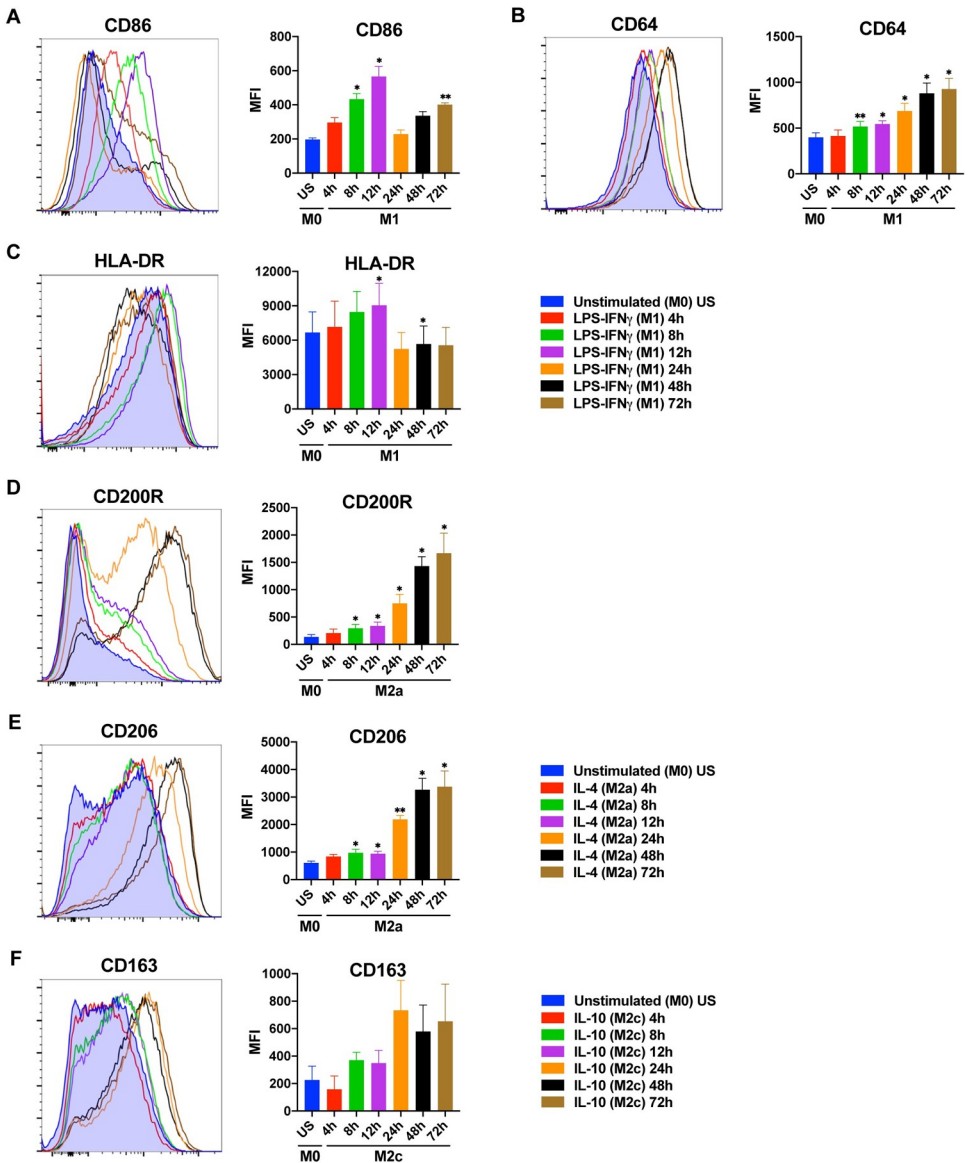

**Fig 2. Time depended changes in the expression of macrophage polarization markers at protein level.** Primary human MDMs (M0) were left unstimulated or were polarized to M1 (100 ng/mL LPS, 20 ng/mL IFNγ) or M2a (20 ng/mL IL-4) for 4, 8, 12, 24, 48, or 72 h. Surface marker expression **(A-C)** of CD86, CD64 and HLA-DR for M1 polarization; **(D-E)** of CD200R and CD206 (MRC1) for M2a polarization, and **(F)** of CD163 for M2c polarization was analyzed by flow cytometry. Shown are representative histograms (left) and summary data shown are mean ± SEM of biological replicates of 3 independent donors (right). Polarized macrophages (M1, M2a, or M2c) at all time points were compared with unstimulated (US) M0 macrophages. Trends for individual donors are depicted in S2 Fig. Statistical analyses were performed with repeated measures ANOVA, * p < 0.05; ** p < 0.01; *** p < 0.001 (Statistical comparison of all time points are shown in the S2 Table).

In M2a polarized macrophages, the expression of CD200R and CD206 (MRC1) was significantly increased starting from 8h of stimulation with the highest levels reached after 48–72 h (Fig 2D and 2E). In M2c polarized macrophages, although the expression of CD163 was increased after 24 h of stimulation, it did not reach statistical significance (Fig 2F). This seems to be due to the variation between donors (S2 Fig).

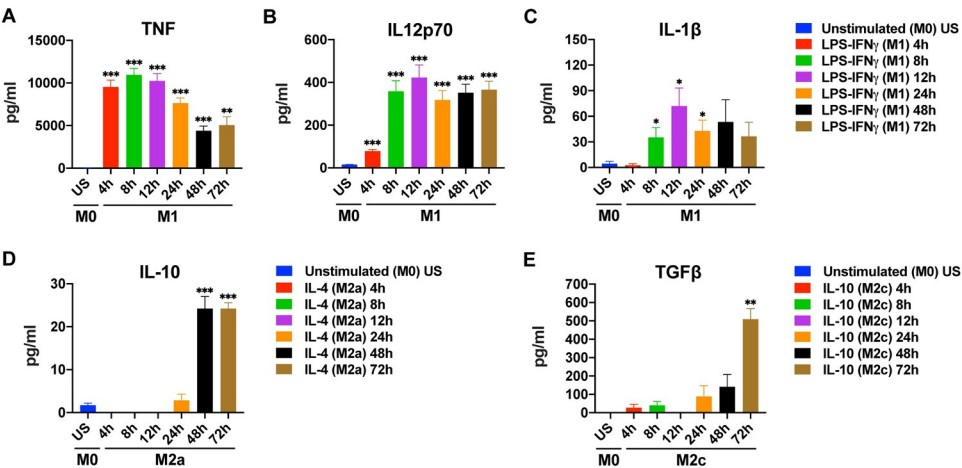

**Fig 3. Time depended changes in cytokine production by M1, M2a, and M2c macrophages.** Primary human MDMs (M0) were left unstimulated or were polarized to M1 (100 ng/mL LPS, 20 ng/mL IFNγ), M2a (20 ng/mL IL-4), or M2c (20 ng/mL IL-10) for 4, 8, 12, 24, 48, or 72 h. Production of **(A-C)** TNF, IL-12 and IL-1β for M1 polarization; of **(D)** IL-10 for M2a polarization, and of **(E)** TGFβ for M2c polarization was analyzed by ELISA. Data shown are mean ± SEM of biological replicates of 3 independent donors. Polarized macrophages (M1, M2a, or M2c) at all time points were compared with unstimulated (US) M0 macrophages. Trends for individual donors are depicted in S3 Fig. Statistical analyses were performed with repeated measures ANOVA, $^*$ $p < 0.05$; $^{**}$ $p < 0.01$; $^{***}$ $p < 0.001$ (Statistical comparison of all time points are shown in the S3 Table).

To test if the M2a and M2c surface markers were specific to these subtypes, the expression of the M2 markers were analyzed in M1, M2a, and M2c macrophages by flow cytometry. The strong upregulation of CD206 on M2a macrophages was not seen on M1 or M2c macrophages. Moreover, the expression of the M2c marker CD163 was not increased on M1 and M2a macrophages (S4 Fig). This confirms that the selected M2 markers were appropriate to identify their respective macrophage subtype.

These data indicate that 12 h of stimulation is best for the M1 surface markers CD86 and HLA-DR, whereas 24–72 h of stimulation is needed for the M1 marker CD64 and the M2 markers CD200R, CD206, and CD163 (Table 1).

### Time course depended changes in cytokine production by M1, M2a and M2c macrophages

Finally, the production of key cytokines by the M1, M2a, or M2c polarized human MDMs was analyzed by ELISA (Fig 3, S3 Fig). TNF production by M1 macrophages peaked at 4, 8, and 12 h of stimulation, with a slight decrease at later time points (Fig 3A). IL-12 production was low at 4 h but prominent and stable between 8 h to 72 h of stimulation (Fig 3B). In contrast, IL-1β production by M1 macrophages was significantly increased between the 8 h-24 h time points (Fig 3C). In M2 macrophages, a significant production of cytokines was only observed after 48 h (IL-10) and 72 h (IL-10, TGFβ) of stimulation (Fig 3D and 3E).

These data indicate that 8–24 h of stimulation is best for the simultanious detection of the M1 cytokines IL-1β, IL-12p70, and TNF, whereas 72 h of stimulation is needed for the detection of the M2 cytokines IL10 and TGFβ, simultaneously (Table 1).

### Discussion

Macrophages can polarize into different functional phenotypes according to the signals they receive from the environment. Environmental signals that induce macrophage polarization in

vivo can be mimicked in in vitro studies [5, 6, 41]. Although the stimulation time affects the expression levels of most markers used to analyze M1 and M2 polarization, there is no consensus in the literature regarding the timing to analyze particular markers during in vitro experiments. Here we report a detailed time course of M1, M2a, and M2c polarization markers of primary human macrophages that determine the optimal time points for their detection at mRNA and protein levels.

Our results demonstrate a dramatic increase in the expression of the chemokines CXCL9 and CXCL10 and the cytokine TNF at mRNA level in the first 4 h of stimulation of primary MDMs polarized into the M1 phenotype with LPS-IFNγ (Fig 1A–1C). In M1 polarization, the upward trend in gene expression decreased with prolonged stimulation, for these genes. Interestingly, in contrast to the other M1 markers tested, IDO1 expression was increased with prolonged stimulation, reaching its highest value between 48 and 72 h (Fig 1F). Since IDO1 is a regulatory enzyme, that is induced after inflammation [42], it is understandable that it is induced later than cytokines and chemokines. Our results are in accordance with previous studies, in which macrophages were stimulated for 48 h to determine IDO1 expression [43, 44]. Based on our results, 4 h LPS-IFNγ stimulation, seems to be the optimum time point to observe the changes in M1 markers at mRNA level. Although IDO1 mRNA expression peaked at late stimulation points, its expression was also quite high at 4 h of stimulation (Fig 1F). In M2a macrophages, CCL17 expression was high at all time points studied, while CCL22 expression was increased from 4 h to 48 h of stimulation (Fig 1I and 1J).

The surface markers CD64, CD86, and HLA-DR are commonly used as M1 polarization markers [19]. In our study, the expression of CD86 and HLA-DR on M1 macrophages showed the strongest increase at the 12 h time point followed by a decrease at 24–72 h (Fig 2A and 2C), similar to previous studies [45]. Stimulation of macrophages with LPS leads to the production IL-10 that inhibits CD86 in an autocrine manner to prevent overexpression of CD86 [46]. Therefore, the observed decrease in CD86 and HLA-DR could be due to the down-regulation of these receptors by IL-10 [46–48]. It has been reported that CD86 expression is higher in both M1 and M2 macrophages than unpolarized MDMs at 48 h [45], and 72 h [13] of stimulation. Based on our results, the optimum stimulation time for CD86 and HLA-DR to be considered as M1 polarization markers is 12 h, and their expression on M1 macrophages decreases thereafter. In contrast, CD64 expression on M1 macrophages was increased from 8 h to 72 h of stimulation (Fig 2B), unlike HLA-DR and CD86 expression. CD64 expression has been reported to be consistently high on M1 macrophages over a wide range of stimulation times [13, 41, 49]. CD64 has been defined as one of the most distinctive M1 polarization markers [41].

While CD206 is a recognized M2a marker for murine macrophages [50], some studies suggested that it was not a specific marker for M2a polarization in humans. [49, 51]. It has been reported that MDMs polarized into M2a phenotype for 24 h, upregulated CD200R but not CD206 [51]. According to our results, the expression of the surface markers CD200R and CD206 gradually increased over time in human MDMs polarized into M2a with IL-4, reaching a maximum level at 48 and 72 h of stimulation (Fig 2D and 2E). Expression of the mannose receptor CD206 (MRC1) at mRNA level also peaked at 72 h of stimulation in M2a macrophages (Fig 1G). Furthermore, the strong upregulation of CD206 on M2a macrophages was not seen on M1 or M2c macrophages, confirming that CD206 is specific to M2a polarization. (S4 Fig).

The scavenger receptor CD163 is a specific M2c polarization marker (Vogel et al., 2014). In our study, the surface CD163 expression on IL-10 stimulated M2c macrophages was increased after 24–72 h of stimulation (Fig 2F), which is consistent with the previous studies [41, 45, 49]. However, it did not reach statistical significance, which seems to be due to donor variability

(S2 Fig). In M2c macrophages, CD163 gene expression was increased at 4 h and onwards (Fig 1M), in accordance with this, CD163 surface marker expression peaked at later hours as expected (Fig 2F).

We also examined the impact of time on the cytokine production and gene expression levels of polarized MDMs. The production of IL-1β, IL12p70, and TNF was high after a wide range of stimulation times in M1 macrophages with 8–12 h of stimulation being optimal (Fig 3A–3C). A previous study has reported TNF and IL12p40 production in M1 macrophages after 48 h of stimulation, while no IL-1β production was detected [41]. In M1 macrophages, IL-12 gene expression was increased at 4 h of stimulation (Fig 1E); which was mirrored by an increase of IL-12 production after 4 h (Fig 3B). The increase in gene expression at 4 h was followed by a decrease below basal level starting from 8 h and another increase after 48 h. It has been shown that the transcription factor IRF5 remains bound to the IL12 promoter for 16 hours after LPS stimulation. The binding peaks at the 4th hour of stimulation and then decreases [52]. Decreased IRF5 binding may account for the fluctuation in IL-12 gene expression in our results. In M1 polarization, the increase of IL-1β gene expression at 4 h of stimulation triggered IL-1β production at 8–24 h of stimulation (Fig 1D). Contrary to the increase in IL-10 gene expression in M2c macrophages (Fig 1L), the gene expression of IL-10 was decreased below basal level in M2a macrophages (Fig 1K). In line with our finding, it was reported that M2a (IL-4) stimulation did not trigger IL-10 gene expression in bone-marrow derived macrophages after 24-hour stimulation [53]. In contrast, however, THP-1 cells displayed enhanced IL-10 expression after 24 hours of M2a (IL-4) stimulation [54]. These differing results may be due to the different response of primary macrophages and cell lines. Although IL-10 gene expression was not increased in M2a macrophages (Fig 1K), IL-10 production was observed after 48 h (Fig 3D). In M2c macrophages, TGFβ gene expression was not detectable at any time of stimulation but prominent TGFβ production was observed at 72 h of stimulation (Fig 3E). Although one would expect that an increase at the protein level follows an increase at mRNA level, this is not always the case, as current mRNA expression could suffice to maintain protein expression [55] or as protein accumulates for later release [56, 57].

In summary, we report here the optimal time points for the analysis of the expression of the main M1, M2a, and M2c polarization markers. It is unfortunate that there is no consensus yet regarding the optimal stimulation times for particular macrophage polarization markers in in vitro experiments, as this can lead to false-negative results and impedes the comparison of different studies. Therefore, our study provides useful guideline for the optimal polarization times of primary human macrophages, which will likely facilitate the investigation of macrophage polarization and its reproducibility.

## Supporting information

**S1 Table. Statistical analyses for M1, M2a and M2c markers at mRNA level in polarized macrophages.** Expression of the indicated markers at the time points shown were compared by repeated measures ANOVA, * $p < 0.05$; ** $p < 0.01$; *** $p < 0.001$. ns, not significant; US, unstimulated.
(PDF)

**S2 Table. Statistical analyses for M1, M2a and M2c surface markers in polarized macrophages.** Expression of the indicated markers at the time points shown were compared by repeated measures ANOVA, * $p < 0.05$; ** $p < 0.01$; *** $p < 0.001$. ns, not significant; US, unstimulated.
(PDF)

**S3 Table. Statistical analyses for M1, M2a and M2c cytokines in polarized macrophages.**
Expression of the indicated markers at the time points shown were compared by repeated
measures ANOVA, * $p < 0.05$; ** $p < 0.01$; *** $p < 0.001$. ns, not significant; US, unstimulated.
(PDF)

**S1 Fig. Time-dependent changes in the expression of macrophage polarization markers at
the mRNA level for each donor.**
(TIF)

**S2 Fig. Time-dependent changes in the expression of macrophage polarization markers at
the protein level for each donor.**
(TIF)

**S3 Fig. Time-dependent changes in cytokine production for each donor.**
(TIF)

**S4 Fig. Time depended changes in the expression of M2 markers in M1, M2a and M2c macrophages analysed by flow cytometry.** Summary data shown are mean ± SEM of biological
replicates of 3 independent donors. Polarized macrophages (M1, M2a, or M2c) at all time
points were compared with unstimulated (US) M0 macrophages. Statistical analyses were performed with repeated measures ANOVA, * $p < 0.05$; ** $p < 0.01$; *** $p < 0.001$.
(TIF)

## Acknowledgments

We thank Dr. Safiye Nese Atabey and Dr. Mehtap Yuksel Egrilmez for valuable scientific contributions. We thank the Flow Cytometry and Cell Sorting Facility at the Izmir Biomedicine
and Genome Center for excellent technical assistance.

## Author Contributions

**Conceptualization:** Duygu Sag.

**Formal analysis:** Duygu Unuvar Purcu, Asli Suner.

**Funding acquisition:** Duygu Sag.

**Investigation:** Duygu Unuvar Purcu, Asli Korkmaz, Sinem Gunalp, Derya Goksu Helvaci,
Yonca Erdal.

**Methodology:** Duygu Unuvar Purcu, Asli Korkmaz, Sinem Gunalp, Derya Goksu Helvaci,
Yonca Erdal, Duygu Sag.

**Project administration:** Duygu Sag.

**Resources:** Yavuz Dogan, Gerhard Wingender.

**Supervision:** Duygu Sag.

**Validation:** Duygu Unuvar Purcu, Duygu Sag.

**Visualization:** Duygu Unuvar Purcu, Asli Korkmaz, Sinem Gunalp, Derya Goksu Helvaci,
Gerhard Wingender, Duygu Sag.

**Writing – original draft:** Duygu Unuvar Purcu, Asli Suner, Gerhard Wingender, Duygu Sag.

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
