## [Decision Letter · Decision Letter 0]

1 Oct 2021

PONE-D-21-23747Effect of stimulation time on the expression of human macrophage polarization markersPLOS ONE

Dear Dr. Sag,

Thank you for submitting your manuscript to PLOS ONE. After careful consideration, we feel that it has merit but does not fully meet PLOS ONE’s publication criteria as it currently stands. Therefore, we invite you to submit a revised version of the manuscript that addresses the points raised during the review process.

The manuscript addresses an important question and results presented are of interest. However, a number of issues have been raised by the Reviewers, and need to be all addressed. In particular, a more comprehensive phenotype of differentiated macrophages should be provided in order to clearly determine the type and level of cell differentiation.

We look forward to receiving your revised manuscript.

Kind regards,

Alain Haziot, M.D.

Academic Editor

PLOS ONE

Journal Requirements:

Reviewers' comments:

Reviewer's Responses to Questions

**Comments to the Author**

1. Is the manuscript technically sound, and do the data support the conclusions?

Reviewer #1: Partly

Reviewer #2: Partly

2. Has the statistical analysis been performed appropriately and rigorously? 

Reviewer #1: I Don't Know

Reviewer #2: No

3. Have the authors made all data underlying the findings in their manuscript fully available?

Reviewer #1: Yes

Reviewer #2: Yes

4. Is the manuscript presented in an intelligible fashion and written in standard English?

Reviewer #1: Yes

Reviewer #2: Yes

5. Review Comments to the Author

Reviewer #1: Purcu and colleagues used primary human monocyte-derived macrophages to determine the optimal time points for analyses of various M1, M2a and M2c polarization markers following in vitro stimulation of MDM.

The authors showed that polarization markers that delineate M1, M2a and M2c MDM were optimally expressed at different times post-stimulation. Results are of interest, addressing important questions regarding the optimal time to use in vitro polarized M1, M2a and M2c MDM. Experiments are generally well-performed. The manuscript is well-written and easy to read, and the figures are generally well-presented. I have one major comment.

Major comment

The figure legends for all 3 figures highlight that the data in each of the figures is derived from 3 donors. The issue of donor-donor variability is widely reported and therefore, the authors should present the data for each donor separately. This would show if “the optimal time points for the analysis of the expression of the main M1, M2a, and M2c polarization markers” is consistent for each donor, and the most convincing way to support the conclusions of the authors.

Reviewer #2: This article was designed by the authors to identify the best time of activation (from 4h to 72h) to characterize specific markers of human monocyte-derived macrophages polarized into M1, M2a and M2c by using RT-PCR for mRNA expression analysis, flow cytometry to characterize membrane specific markers and ELISA to quantify secretion of cytokines/chemokines.

The data are informative but are not enough ambitious.

Determine the best time of polarization is necessary and the authors did it but I suggest adding more markers per polarization profile. In fact, it will be interesting to have the data for the same markers at the mRNA and secreted markers ; It was done only for TNFa in M1 MDM.

Moreover, some markers are not so specific to a type of polarization (CD206 for example) that is why it will be informative to have the expression in both the 3 phenotypes. In fact, to characterize a population of MDM from patients for example, we need to be confident with the difference of expression between subtypes.

The authors should modulate their conclusion, because the most important thing is to know if a marker is up or not and not so much how fold increased it is, except if we compared it to an another polarization profile ; that was not done here. In fcat, is more important to known when the marker is not expressed than is significantly expressed but lower.

I suggest to precise the % of viability cells after 24h, 48h and 72H, as LPS at 100 ng/ml may be strong for M1 MDM. These data can be extracted form flow cytometry analysis since a fixable viability kit was used. The pic of isotype control should appear on flow cytometry graph.

Student’t test cannot be used for the figures but an Anova test.

Please corrected page 10, paragraph 3.3 : “analyzed by flow cytometry” replace by “analyzed by ELISA”.

Please precise the number of replicate, it is not clear : if data are representative of 2 experiments, statistics cannot be done.

6. PLOS authors have the option to publish the peer review history of their article (what does this mean?). If published, this will include your full peer review and any attached files.

Reviewer #1: No

Reviewer #2: No

---

## [Author Response · Author response to Decision Letter 0]

30 Jan 2022

Response to the Reviewers, PLOS ONE

We were pleased to find that the reviewers and the editors thought that our manuscript entitled “Effect of stimulation time on the expression of human macrophage polarization markers” by Purcu et al., was of interest and we thank you for the positive comments that we received and for giving the opportunity to submit a revised version. We also appreciate the editor’s and the reviewers' time and effort in providing valuable feedback that has helped us enhance the manuscript. 

Please find the revised manuscript attached, which incorporates the reviewers' and editor's suggestions and highlights the significant changes in the text in yellow. Here is our point-by-point response to the concerns raised.

We believe that we have adequately addressed all the concerns raised by the academic editor and the reviewers, and hope that our revised manuscript is now suitable for publication in PLOS ONE.

Academic Editor:

The manuscript addresses an important question and results presented are of interest. However, a number of issues have been raised by the Reviewers, and need to be all addressed. In particular, a more comprehensive phenotype of differentiated macrophages should be provided in order to clearly determine the type and level of cell differentiation.

Response: We thank the editor for this valuable comment. We would like to emphasize that our goal was to determine the optimum stimulation time for the expression of established M1/M2 polarization markers, rather than to specify the type and level of cell differentiation. We believe, we incorporated the most important and widely used polarization markers for each macrophage cell type. However, as the reviewers suggested, all the secreted markers that had been previously analyzed by ELISA were also analyzed at mRNA level by qPCR. The data for “IL1β and IL12A” as M1 Markers; “IL10” as M2a marker; “TGFβ” as M2c marker were incorporated in the new Fig 1 (D, E, K and N). The Results and Discussion sections were modified accordingly. 

Reviewer #1: 

Purcu and colleagues used primary human monocyte-derived macrophages to determine the optimal time points for analyses of various M1, M2a, and M2c polarization markers following in vitro stimulation of MDM.

The authors showed that polarization markers that delineate M1, M2a, and M2c MDM were optimally expressed at different times post-stimulation. Results are of interest, addressing important questions regarding the optimal time to use in vitro polarized M1, M2a, and M2c MDM. Experiments are generally well-performed. The manuscript is well-written and easy to read, and the figures are generally well-presented. I have one major comment.

Major comment

1- The figure legends for all 3 figures highlight that the data in each of the figures are derived from 3 donors. The issue of donor-donor variability is widely reported and therefore, the authors should present the data for each donor separately. This would show if “the optimal time points for the analysis of the expression of the main M1, M2a, and M2c polarization markers” is consistent for each donor, and the most convincing way to support the conclusions of the authors.

Response 1

We thank the reviewer for this excellent suggestion. We incorporated the new supplementary figures (S1 Fig, S2 Fig and S3 Fig) that shows for each donor separately the changes over time of the expression of the analysed M1, M2a, and M2c polarization markers. This allows now to evaluate the donor-to-donor variability for each marker and each time point.

Reviewer #2: 

This article was designed by the authors to identify the best time of activation (from 4h to 72h) to characterize specific markers of human monocyte-derived macrophages polarized into M1, M2a, and M2c by using RT-PCR for mRNA expression analysis, flow cytometry to characterize membrane specific markers and ELISA to quantify secretion of cytokines/chemokines.

The data are informative but are not enough ambitious.

1- Determine the best time of polarization is necessary and the authors did it but I suggest adding more markers per polarization profile. In fact, it will be interesting to have the data for the same markers at the mRNA and secreted markers; It was done only for TNF in M1 MDM.

Response 1

We thank the reviewer for this excellent suggestion. In this revised manuscript, all the secreted markers that had been previously analyzed only by ELISA were now also analyzed at mRNA level by qPCR. The data for “IL1β and IL12A” as M1 Markers; “IL10” as M2a marker; “TGFβ” as M2c marker were incorporated in the new Fig 1 (D, E, K and N). The Results and Discussion sections were modified accordingly. 

2- Moreover, some markers are not so specific to a type of polarization (CD206 for example) that is why it will be informative to have the expression in both the 3 phenotypes. In fact, to characterize a population of MDM from patients, for example, we need to be confident with the difference of expression between subtypes. 

Response 2

We thank the reviewer for this valuable recommendation. As suggested, the expression of the M2 markers were analyzed in each macrophage subtypes by flow cytometry. In accordance with the literature (Stein et al. 1992; Gordon 2003; Mantovani et al. 2004, 2005; Davis et al. 2013; Murray et al. 2014), the strong upregulation of CD206 on M2a macrophages was not seen on M1 or M2c macrophages. Likewise, the expression of the M2c marker “CD163” was not increased on M1 or M2a macrophages. Therefore, we conclude that the M2 markers we chose for each macrophage subtype were appropriate. We incorporated these data in the new supplementary figure (S4 Fig) and modified the Results section accordingly.

3- The authors should modulate their conclusion, because the most important thing is to know if a marker is up or not and not so much how fold increased it is, except if we compared it to another polarization profile; that was not done here. In fact, is more important to know when the marker is not expressed than is significantly expressed but lower. 

We thank the reviewer for this comment. We agree with the reviewer that the ability to detect a polarization marker is essential to define the macrophage subsets. The data on significant differences in expression are provided throughout our manuscript. Obviously, the stronger a marker is expressed, the easier it is to identify a given subset. Therefore, providing the fold-changes offers the readers the possibility to select the optimal time point for their experimental needs. As supporting the experimental design of our readers is the main goal of our manuscript we feel that taking this aspect out would weaken the usability of our data for the readers. However, we understand the reviewer’s concern and modified our Table 1 to specify a range for the suggested stimulation times for all markers. 

4- I suggest to precise the % of viable cells after 24h, 48h, and 72H, as LPS at 100 ng/ml may be strong for M1 MDM. These data can be extracted from flow cytometry analysis since a fixable viability kit was used. 

Response 4

We thank the reviewer for this comment. As the reviewer suggested, we re-analyzed our flow cytometry data to determine the percentages of the viable cells after 100 ng/ml LPS + 10 ng/ml IFN� treatment. Statistical analysis showed that there is no significant difference between the 100 ng/ml LPS + 10 ng/ml IFN� treated and untreated cells at 24, 48 and 72 h, as shown below. We incorporated this information in the “Polarization of human MDMs” part of the Materials and Methods section as “The viability of the LPS+ IFN� treated human MDMs was analyzed by flow cytometry and was over 92% for all time points.” 

5- The pic of isotype control should appear on the flow cytometry graph.

Response 5

We thank the reviewer for this comment. We view isotype controls critically and follow other experts in the field with a similar opinion. Investigators chose, among several antibody options, usually the isotype control that yields the lowest background, basically obliterating by this selection the indented effect.

“Isotype controls have a long history in flow cytometry and are meant to account for nonspecific staining of an antibody of a particular isotype conjugated to a particular fluorochrome. However, even when the control antibody is isotype matched to the test antibody, there are two main limitations to the usefulness of this type of control. The first limitation is that individual antibody conjugates have various levels of background staining, depending upon their specificity, concentration, degree of aggregation, and fluorophore: antibody ratio, among other variables. It is thus a hit-or-miss prospect to find an isotype control that truly matches the background staining of a particular test antibody. And, remembering that we are using the isotype control to help us define the true level of background staining, this becomes a circular proposition.” (Maeker & Trotter, Cytometry 2006, PMID: 16888771)

Similar to Maeker & Trotter, we do believe that biological controls are preferable. 

“For example, in stimulation assays, the unstimulated (or irrelevantly stimulated) sample usually provides the best means to distinguish positive from negative events” (Maeker & Trotter, Cytometry 2006, PMID: 16888771)

Isotype controls are often used in cases were such a biological control might be misleading. Therefore, we feel confident that using unstimulated cells as negative control is a valid and stringent control in this manuscript.

6- Student’t test cannot be used for the figures but an Anova test. 

Response 6

As the reviewer suggested, statistical analyses were performed with Anova test. The Methods section and the figure legends were modified accordingly in this revised manuscript. The analysis was performed by a bioistatician, who is now added as a coauthor to the revised manuscript. 

7- Please correct page 10, paragraph 3.3 : “analyzed by flow cytometry” replace by “analyzed by ELISA”.

Response 7

We apologize for this omission. Page 14, paragraph 4.3: “analyzed by flow cytometry” is replaced by the correct phrase “analyzed by ELISA” at lane 301. We thank the reviewer for their attention. 

8- Please precise the number of replicate, it is not clear : if data are representative of 2 experiments, statistics cannot be done. 

Response 8

We apologize for the unclear phrasing. In the revised manuscript, we corrected the phrase as “Data shown are mean ± SEM of biological replicates of 3 independent donors” in both the Methods section and the figure legends. We thank the reviewer for pointing this out to us. 

REFERENCES:

Davis MJ, Tsang TM, Qiu Y, Dayrit JK, Freij JB, Huffnagle GB, et al. Macrophage M1/M2 polarization dynamically adapts to changes in cytokine microenvironments in Cryptococcus neoformans infection. MBio [Internet]. 2013 [cited 2021 Oct 25];4(3). Available from: https://journals.asm.org/journal/mbio

Gordon S. Alternative activation of macrophages [Internet]. Vol. 3, Nature Reviews Immunology. Nature Publishing Group; 2003 [cited 2020 Aug 19]. p. 23–35. Available from: https://www.nature.com/articles/nri978

Mantovani A, Sica A, Locati M. Macrophage polarization comes of age. Vol. 23, Immunity. 2005. p. 344–6. 

Mantovani A, Sica A, Sozzani S, Allavena P, Vecchi A, Locati M. The chemokine system in diverse forms of macrophage activation and polarization [Internet]. Vol. 25, Trends in Immunology. Oxford University Press, Oxford; 2004 [cited 2017 Sep 7]. p. 677–86. Available from: http://www.ncbi.nlm.nih.gov/pubmed/15530839

Murray PJ, Allen JE, Biswas SK, Fisher EA, Gilroy DW, Goerdt S, et al. Macrophage Activation and Polarization: Nomenclature and Experimental Guidelines. Immunity [Internet]. 2014 Aug 17 [cited 2020 Aug 26];41(1):14–20. Available from: http://dx.doi.org/10.1016/j.immuni.2014.06.008

Stein M, Keshav S, Harris N, Gordon S. Interleukin 4 potently enhances murine macrophage mannose receptor activity: a marker of alternative immunologic macrophage activation. J Exp Med [Internet]. 1992 Jul 1 [cited 2021 Oct 25];176(1):287–92. Available from: http://rupress.org/jem/article-pdf/176/1/287/1102739/287.pdf

---

## [Decision Letter · Decision Letter 1]

22 Feb 2022

PONE-D-21-23747R1Effect of stimulation time on the expression of human macrophage polarization markersPLOS ONE

Dear Dr. Sag,

Thank you for submitting your manuscript to PLOS ONE. After careful consideration, we feel that it has merit but does not fully meet PLOS ONE’s publication criteria as it currently stands. Therefore, we invite you to submit a revised version of the manuscript that addresses the points raised during the review process.

A couple of minor comments need to be addressed. Your revised manuscript would then go through an expedited review.

We look forward to receiving your revised manuscript.

Kind regards,

Alain Haziot, M.D.

Academic Editor

PLOS ONE

Journal Requirements:

Reviewers' comments:

Reviewer's Responses to Questions

**Comments to the Author**

1. If the authors have adequately addressed your comments raised in a previous round of review and you feel that this manuscript is now acceptable for publication, you may indicate that here to bypass the “Comments to the Author” section, enter your conflict of interest statement in the “Confidential to Editor” section, and submit your "Accept" recommendation.

Reviewer #1: All comments have been addressed

Reviewer #2: All comments have been addressed

2. Is the manuscript technically sound, and do the data support the conclusions?

Reviewer #1: Yes

Reviewer #2: Yes

3. Has the statistical analysis been performed appropriately and rigorously? 

Reviewer #1: Yes

Reviewer #2: Yes

4. Have the authors made all data underlying the findings in their manuscript fully available?

Reviewer #1: Yes

Reviewer #2: Yes

5. Is the manuscript presented in an intelligible fashion and written in standard English?

Reviewer #1: Yes

Reviewer #2: Yes

6. Review Comments to the Author

Reviewer #1: (No Response)

Reviewer #2: The authors have addressed my recommendations

I have some minors recommendations:

- I suggest to add in the table 1 the new data on Il-10 and on TGF-b mRNA levels in M2a and M2c macrophages respectively

- see the reference of Actin B primers lines 168 an 173.

7. PLOS authors have the option to publish the peer review history of their article (what does this mean?). If published, this will include your full peer review and any attached files.

Reviewer #1: No

Reviewer #2: No

---

## [Author Response · Author response to Decision Letter 1]

24 Feb 2022

Response to the Reviewers, PLOS ONE

We were pleased to find that the reviewers and the editors thought that our manuscript entitled “Effect of stimulation time on the expression of human macrophage polarization markers” by Purcu et al., was of interest and that we addressed all of their concerns raised in our revised manuscript. We thank you for giving us the opportunity to submit a revised version. We also appreciate the editor’s and the reviewers' time and effort in providing valuable feedback that has helped us to enhance the manuscript. 

Please find the revised manuscript attached, which incorporates the reviewers’ minor comments and highlights the significant changes in the text in yellow. Please find our point-by-point response to the comments below.

We believe that we have adequately addressed all the concerns raised by the reviewers, and hope that our revised manuscript is now suitable for publication in PLOS ONE.

Reviewer #2: 

1- I suggest to add in the table 1 the new data on Il-10 and on TGF-b mRNA levels in M2a and M2c macrophages respectively

Response 1

We thank the reviewer for this comment. We added information on IL-10 in M2a macrophages and TGFβ in M2c macrophages at mRNA level to the revised Table 1.

2- See the reference of Actin B primers lines 168 an 173.

Response 2

We thank the reviewer for this comment and apologize for the confusion. The Actin B primers mentioned in lines 168 and 173 have different reference ID’s as they are primers from different manufacturers. In the first version of the manuscript, all the real time PCR experiments were performed using the RealTime ready Single Assays by Roche. However, as these assays were later discontinued, for the revision we had to use a different assay with a different reference ID.

---

## [Editor Report · Decision Letter 2]

28 Feb 2022

Effect of stimulation time on the expression of human macrophage polarization markers

PONE-D-21-23747R2

Dear Dr. Sag,

We’re pleased to inform you that your manuscript has been judged scientifically suitable for publication and will be formally accepted for publication once it meets all outstanding technical requirements.

Kind regards,

Alain Haziot, M.D.

Academic Editor

PLOS ONE
---

## [Editor Report · Acceptance letter]

4 Mar 2022

PONE-D-21-23747R2 

Effect of stimulation time on the expression of human macrophage polarization markers 

Dear Dr. Sag:

I'm pleased to inform you that your manuscript has been deemed suitable for publication in PLOS ONE. Congratulations! Your manuscript is now with our production department. 

Kind regards, 

on behalf of

Dr. Alain Haziot 

Academic Editor

PLOS ONE